# Factors Associated with Attitudes toward Aging among Taiwanese Middle-Aged and Older Adults: Based on Population-Representative National Data

**DOI:** 10.3390/ijerph19052654

**Published:** 2022-02-24

**Authors:** Shu-Hsin Lee, Chih-Jung Yeh, Cheng-Yu Yang, Ching-Yi Wang, Meng-Chih Lee

**Affiliations:** 1Department of Nursing, Chung Shan Medical University, Taichung 402306, Taiwan; shl@csmu.edu.tw; 2Department of Nursing, Chung Shan Medical University Hospital, Taichung 402306, Taiwan; 3Institute of Medicine, Chung Shan Medical University, Taichung 402306, Taiwan; 4Department of Public Health, Chung Shan Medical University, Taichung 402306, Taiwan; alexyeh@csmu.edu.tw; 5Department of Physical Therapy, Chung Shan Medical University, Taichung 402306, Taiwan; clack088688@gmail.com; 6Physical Therapy Room, Chung Shan Medical University Hospital, Taichung 402306, Taiwan; 7Department of Family Medicine, Taichung Hospital, Ministry of Health and Welfare, Taichung 403301, Taiwan; 8Institute of Population Health Sciences, National Health Research Institutes, Miaoli 350401, Taiwan; 9College of Management, Chaoyang University of Technology, Taichung 413310, Taiwan

**Keywords:** attitude toward aging, leisure activities, physical exercise, volunteer service

## Abstract

In middle-aged and older adults, attitude toward aging (ATA) exerts significant influences on their current and future health. For health promotion to be successful, participants’ ATA requires health care providers’ attention. Knowing the factors associated with ATA can facilitate future studies to investigate effective interventions. The aim of this study was to identify the factors associated with ATA in middle-aged and older adults. This cross-sectional study analyzed data of a nationally representative sample of adults aged 58 years and older collected in a population-based longitudinal study: the Taiwan Longitudinal Study on Aging (TLSA). To identify the factors associated with ATA, we investigated demographic factors (age, gender, education, marital and cohabitation status, and financial satisfaction status) and health-related factors (number of co-morbidities, depression, physical function dependency) with bivariate analysis and multiple regression analysis. To identify the activities beneficial to ATA over and above demographic and health-related factors, various activities (exercise, volunteer service, and leisure activities) were each examined individually by multiple regression analysis. The factors detrimental to ATA were advanced age, a higher number of co-morbidities, living alone, depression, and dependence on physical function. Those beneficial to ATA were higher education, financial satisfaction, physical exercise, volunteer service, and six leisure activities.

## 1. Introduction

Attitude toward aging (ATA) refers to one’s beliefs about aging and older people in general and it is gradually built up through a process of assimilation from one’s surrounding culture and internalized self-relevant attitudes and stereotypes across one’s life span [1]. One’s ATA significantly influences one’s health. In the literature, ATA is often measured by questionnaires such as Attitudes toward own Aging subscale of the Philadelphia Geriatric Center Morale Scale (PGCMS) [2], Attitudes to Aging Questionnaire (AAQ) [3], and Expectations Regarding Aging (ERA-12) [4]. Studies consistently show that negative ATA, assessed by the above mentioned questionnaires, is associated with poorer health, less control over one’s lifestyle behaviors, poorer functioning performance, and lower life satisfaction [5,6,7,8]. More importantly, this relationship is found not only in cross-sectional studies but also in longitudinal investigations, and it exists in both middle-aged and older adults [9,10,11,12]. People with negative ATA are less likely to engage in an active lifestyle, participate in activities, or exercise [13]. For health promotion programs to be effective, health care providers need to pay attention to participants’ ATA. Identifying the factors associated with ATA is an important first step prior to the exploration of possible interventions.

Literature has reported inconsistent results on the association of demographic factors (age, sex, education, marriage and living status, income) with ATA [12,13,14,15]. Negative ATA is often associated with many chronic diseases such as hypertension, diabetes and coronary heart disease [11,16], respiratory disease [17], arthritis, asthma, and depression [13,18]. In addition, participating in leisure activities [19], regular exercise [20,21], and volunteer service [22] has been shown to reduce the relationship between ATA and health. Most of the past studies were conducted in western countries, and this issue has been rarely investigated in an Asian population. Since ATA may be affected by living environment and societal culture, the associated factors of ATA in an Asian population needs to be explored. This issue is urgent to be answered especially as Taiwan experiences the fastest rate of aging in the world; the proportion of people over 65 was 8.6% in 2000 and is expected to rise to 20% by 2027 [23,24].

The purposes of this study were (1) to identify specific demographic and health-related factors associated with ATA, and (2) to identify the specific activities that are significantly associated with ATA over and above the demographic and health-related factors in a group of middle-aged and older adults from analysis of population-representative national data.

## 2. Materials and Methods

The Taiwan Longitudinal Study on Aging (TLSA) is a population-based longitudinal study of a nationally representative random sample of adults aged 58 years and older that begun in 1989 and followed up every 3 to 4 years. The details and design of the TLSA have been described elsewhere [25,26].

Data on participants collected in 2011 (TLSA 2011) were used in this cross-sectional study since this was the first year the TAAQ-SF questionnaire was used in this survey to measure ATA. We selected responses without missing data on the Taiwanese Attitude toward Aging Questionnaire Short Form (TAAQ-SF) [27]. To ensure the validity of responses on the TAAQ-SF from the study participants, we further selected those who were cognitively intact, as indicated by a score of ≥6 on the Short Portable Mental Status Questionnaire (SPMSQ) [28]. The current study was approved by the Institutional Review Board of the Health Promotion Administration of the Ministry of Health and Welfare (Approval No. 5 BHP-2007-002). Before recruitment, all participants received a proper explanation of the study and provided informed consent for inclusion in the study. Participants who could read and write signed the written consent documents; those who could not read or write impressed their name stamps or fingerprints with the assistance of family members. The TLSA 2011 variables used in this study were as follows.

### 2.1. Demographic Variables

The participants’ demographic characteristics consisted of age, gender, education (≤6 years, 7–12 years, and ≥13 years), marital/cohabitation status (living with or without spouse/partner), and financial satisfaction (satisfied, average, and dissatisfied). 

### 2.2. Attitude toward Aging (ATA)

Participants’ ATA were indicated by the total score of the TAAQ-SF that consists of eleven items: (1) you feel that you are old, (2) you feel you have time to pursue your own interests, (3) you feel it is a privilege to grow old, (4) you feel that you cannot take care of yourself, (5) you feel that you could help your family, (6) you feel that you are frail, (7) you feel that you are pleasant and joyful, (8) you feel your mind is not clear, (9) you feel you are kind and warm, (10) you feel unsafe, and (11) you feel satisfied with your current life. Each item was rated on a 5-point Likert scale from 1 (strongly disagree) to 5 (strongly agree). Before summing up the total score, items one, four, six, eight, and ten were reverse-scored to make all the items measure a positive ATA. The total score ranged from 11 to 55, with a higher score indicating more positive ATA. The psychometric properties of TAAQ-SF was examined and confirmed in a previous study [27].

### 2.3. Health-Related Variables

#### 2.3.1. Depression

Participants’ depressive symptoms were assessed with the 10-item short form of the Center for Epidemiological Studies Depression Scale (CESD-10) [29]. The total score range is from 0 to 30, with a higher score indicating greater severity of depressive symptoms. Participants were assigned to either the depressed group (CESD-10 ≥ 10) or the non-depressed group (CESD-10 ≤ 9) [30].

#### 2.3.2. Physical Function

Participants’ physical function was assessed with three scales: mobility (lift/carry 12 kg, walk 200 to 300 m, and climb stairs to second or third floor), instrumental activities of daily living (IADL) (shopping for personal items, ability to handle finances, mode of transportation, heavy housework, housekeeping, ability to use telephone, food preparation, responsibility for own medications, and laundry), and activities of daily living (ADL) (bathing, dressing, feeding, transfer, mobility, and toilet use). An individual was considered “independent” if the participant answered, “without difficulty” on all items on the scale; otherwise, as “dependent”. Physical function was further categorized into four groups: independent on all 3 scales, dependent on any 1 scale, dependent on any 2 scales, and dependent on all 3 scales.

#### 2.3.3. Comorbidity

Total numbers of co-morbidities were summed from the following 15 comorbidities: hypertension, diabetes, cardiovascular disease, stroke (or transient ischemic attack), cancer/malignant tumor, chronic obstructive pulmonary disease (COPD), asthma, arthritis, hepatobiliary disease, hip fracture, cataract, glaucoma, chronic kidney disease, gout, and hyperlipidemia.

### 2.4. Activities

#### 2.4.1. Physical Exercise

Participants were grouped based on exercise frequency (no exercise, <3 or ≥3 times per week), exercise time (<30 or ≥30 min per session), and exercise intensity (do not sweat, sweat a little, or sweat a lot). Those who did not exercise were in the “no exercise” group. Those who exercised ≥3 times per week, ≥30 min each time, and sweated a little or a lot during exercise were in the “regular exercise” group, whereas all others were in the “irregular exercise” group.

#### 2.4.2. Social Service (Volunteer) Work and Leisure Activities

Participants were asked if they participated in social service (volunteer) work, and any of the following leisure activities: listening to music/radio, reading newspapers/magazines/books, going online (not interacting with other persons), going online (interacting with other persons), chatting with relatives/friends, playing chess or cards, gardening (horticulture), taking a walk, bicycling, engaging in outdoor fitness activities (such as jogging, playing ball games, hiking, mountain climbing, etc.), and group activities (such as singing, dancing, Tai Chi, Wai-Tan-Kung, etc.).

### 2.5. Data Analysis

Data were analyzed in SPSS 17.0 (2008 SPSS, Inc., Chicago, IL, USA). Participants’ characteristics are reported as descriptive statistics. The associations between the dependent variable (ATA) and categorical independent variables (demographic variables and health-related variables) were examined with one-way analysis of variance (ANOVA). If global ANOVA showed significant results, a post hoc analysis was performed. The effect size (Eta squared, η^2^) was reported for those who showed significant association. The η^2^ was calculated as $\frac{\mathrm{sum}\;\mathrm{of}\;\mathrm{squares}\;\mathrm{between}\;\mathrm{groups}}{\mathrm{total}\;\mathrm{sum}\;\mathrm{of}\;\mathrm{squares}}$, where 0.01 was small, 0.06 was medium, and 0.14 was a large effect [31]. The association of ATA with the continuous independent variable (total number of co-morbidities) was examined with Spearman’s correlation coefficient (r_s_). A correlation coefficient of 0.10–0.29 was considered small; 0.30–0.49, medium, and 0.50–1.0, large [31].

To identify the demographic and health-related variables that were significantly associated with ATA, those which showed significant associations with the dependent variable (ATA) in the bivariate analysis were entered into the multiple regression analysis. To identify the various activities associated with ATA, a hierarchical multiple regression analysis was performed for each activity category, with demographic and health-related variables controlled for. The standardized Beta (β), t statistics, and p value were reported for comparison of the contributions among the factors.

## 3. Results

### 3.1. Sample Characteristics

The 2011 TLSA had 3727 respondents aged 58 to 101 years who completed the interview. In this study, we selected those without missing data on any of the TAAQ-SF items (n = 3272) and without impaired cognition (SPMSQ ≥ 6) (n = 3054). Thus, a final total of 3054 respondents’ data was used for analysis in this study. 

The characteristics of our participants are reported in Table 1. The participants had a mean age of 69.1 (SD = 9.0). Most were men (50.5%) with ≤6 years of education (62.6%) who were living with a spouse/partner (71.9%) and were satisfied with their financial condition (43.4%). On average, our participants had 2.2 co-morbidities, 39.1% were dependent on at least one physical function scale, and 15.3% were depressed. Just under one third (31.5%) did not exercise at all, and 21.3% participated in volunteer service. The top five leisure activities were chatting with friends/relatives (73.4%), taking a walk (67.1%), reading newspapers/magazines (47.9%), listening to music/radio (45.1%), and gardening (horticulture) (41.9%).

### 3.2. Results of Bivariate Analysis

The dependent variable was ATA (TAAQ-SF), and the independent variables were the demographic and health-related factors. The results of ANOVA indicated that all independent variables were significantly associated with ATA (Table 2). Depression (η^2^ = 0.27) and physical function (η^2^ = 0.26) showed large effect sizes. Financial satisfaction (η^2^ = 0.12), age (η^2^ = 0.10), and education (η^2^ = 0.06) showed medium effect sizes. The total number of co-morbidities showed a medium relationship with ATA (r_s_ = −0.315, *p* < 0.001). 

### 3.3. Results of Multiple Regression Analysis: Factors Associated with ATA

All of the independent variables, except gender, were significantly associated with ATA (Table 3). The total variance explained by the model was 46.7% (adjusted R^2^ = 46.4%, F _(14, 2973)_ = 185.9036, *p* < 0.001). In the model, depression (β = −0.32, *p* < 0.001), financial satisfaction (β = 0.28, *p* < 0.001), and dependent on all 3 physical function scales (β = −0.23, *p* < 0.001) showed the largest contributions to the model. 

### 3.4. Activities Associated with ATA

Both physical exercise and volunteering, each individually, were found significant to ATA controlled for the demographic and health-related factors (Table 4). Both regular and irregular exercise, compared with no exercise, showed significant beneficial effects on ATA. Regular exercise (β = 0.10, *p* < 0.001) showed a larger effect on ATA than did irregular exercise (β = 0.05, *p* < 0.01) and volunteer service (β = 0.07, *p* < 0.001). As to the leisure activities, the following six activities showed significant influences on the ATA: gardening (β = 0.08, *p* < 0.001), outdoor fitness activities (β = 0.07, *p* < 0.001), group activities (β = 0.06, *p* < 0.001), reading newspapers/magazines/books (β = 0.05, *p* = 0.01), chatting with friends/relatives (β = 0.04, *p* < 0.01), and listening to radio/music (β = 0.03, *p* < 0.05). 

## 4. Discussion

The purposes of this study were to identify the factors associated with ATA and the activities beneficial to ATA after the identified associated factors being controlled for in the middle-aged and older Taiwanese population. The results of this study revealed that higher education, living with others, and financial satisfaction are protective factors to ATA, whereas advanced age, higher numbers of co-morbidities, depressive mood, and physical functioning dependency were risk factors to ATA. Beyond the influence of demographic and health-related factors, participation of physical exercise, volunteer service, and six leisure activities were all beneficial to ATA in the middle-aged to older adults. 

Among the identified factors associated with ATA, the only non-modifiable factor was advanced age. Consistent with previous studies, older adults’ views on aging often reflect societal beliefs and stereotypes about aging, which primarily convey the negative aspects of growing old [15,32]. Higher education, living with others, satisfaction with financial status, absence of depression, independent physical functions (in mobility, IADL, and AD), and fewer co-morbidities were modifiable beneficial factors to ATA and are worthy of consideration for government policy makers in the development of strategies for enhancing ATA for future middle-aged to older generations. Consistent with the literature, higher education was associated with positive ATA [13,14]. Specifically, people with higher education experienced significantly less psychosocial loss associated with increasing age than those with lower education, measured by Attitude toward Aging Questionnaire (AAQ) [13]. Living alone and unsatisfactory financial condition were associated with negative ATA [12]. In terms of marriage status, those who never married showed significantly less psychological growth as measured by AAQ whereas those who were separated/divorced and widowed also showed less, but not significant, on psychological growth (positive aspect of wisdom and generativity that adults can feel as they grow older) [13].

The influence of gender on ATA was found to be inconsistent from previous studies, in which some reported no influence [12,13,14] whereas others found men showed more positive ATA than women [15]. Our results found the influence of gender on ATA was confounded by education. Gender was significantly associated with ATA in bivariate analysis, in which men showed significantly more positive attitudes than women. This finding was consistent with a previous study [15]. However, gender was not a significant factor with ATA in the multiple regression analysis because it was also significantly associated with education. Thus, when education was entered into the model, gender did not make a significant contribution to it.

Physical exercise and volunteer service were found to be beneficial to ATA over and above the demographic and health-related factors. Compared to the no exercise group, both regular and irregular physical exercise groups showed significant positive influence on ATA, with the regular exercise group showing greater influence than the irregular exercise group. The benefit of participating in volunteer service to ATA might be through the following mechanisms: more opportunities for social interactions, increased sense of self-worth, greater sense of accomplishment, and better self-rated health [22,33].

Among the leisure activities investigated, six activities showed the significant influential effects on ATA. To the best of our understanding, this is the first study to explore the specific activities which have significant impact on the ATA over and above the other influential factors. Our results revealed that gardening was the most influential activity to ATA. Gardening has been reported to improve survival rates in those with mobility limitations or depressed people aged ≥50 years [33]. Both the outdoor fitness and group activities were considered physical leisure activities. These activities are known to be effective in improving physical function (cardiopulmonary, musculoskeletal), mental health, and self-rated health [34,35]. Reading newspapers/magazines/books and listening to radio/music were considered as cognitive leisure activities [36]. Cognitive leisure activities provide intellectual stimulation and thus are beneficial to cognitive function, maintaining motivation, and relieving psychological stress [36]. Chatting with friends/relatives provides important social support.

To the best of our knowledge, this is the first study to examine the factors associated with ATA and to explore the activities associated with ATA while controlling for demographic and health-related factors in a Taiwanese population. The strength of this study is that the results are based on a nationally representative dataset. However, readers need to be aware of the following when generalizing the results. First, these results were based on a nationally representative Taiwanese population and we included those who were cognitively intact (SPMSQ ≥ 6). These results can be generalized only to similar populations. Second, the causal relationships of the identified factors that are beneficial and detrimental to ATA could not be ascertained in this cross-sectional study. Future studies are recommended to further examining the effects of these identified factors on ATA and health, especially the effects of interventions featuring participation in the identified activities on negative ATA. Further investigations focused on different subgroups by gender, financial status, living alone or not, marital status, and living in larger/smaller cities are also recommended.

## 5. Conclusions

Advanced age, a higher number of co-morbidities, living alone, depression, and dependence on physical function were detrimental to ATA. Higher education, financial satisfaction, and participation in physical exercise, volunteer service, and six leisure activities were all beneficial to ATA. The identified beneficial and detrimental factors provide valuable information for future studies and for developing health promotion interventions.

## Figures and Tables

**Table 1 ijerph-19-02654-t001:** Participant characteristics (n = 3054).

Social Demographic Characteristics	Mean ± SD	n (%)
Age (years)	69.1 ± 9.0	
Gender		
Men		1543 (50.5%)
Women		1511 (49.5%)
Education		
≤6 years		1912 (62.6%)
7–12 years		803 (26.3%)
≥13 years		339 (11.1%)
Marital/cohabitation status, (n = 3008)		
Without spouse/partner		844 (28.1%)
With spouse/partner		2164 (71.9%)
Financial satisfaction		
Dissatisfied		564 (18.5%)
Acceptable		1165 (38.2%)
Satisfied		1324 (43.4%)
Attitude toward aging (TAAQ-SF) (scores)	40.6 ± 6.0	
**Health-related variables**		
Depression (CESD-10) (n = 3052)		
Depressed (CESD-10 ≥ 10)		467 (15.3%)
Physical function (n = 3045)		
All 3 scales dependent		190 (6.2%)
Any 2 scales dependent		526 (17.3%)
Any 1 scale dependent		474 (15.6%)
All 3 scales independent		1855 (60.9%)
Numbers of co-morbidities	2.2 ± 1.8	
**Activities**		
Physical exercise (n = 2996)		
No exercise		944 (31.5%)
Irregular exercise		989 (33.0%)
Regular exercise		1063 (35.5%)
Volunteer service (yes)		454 (14.9%)
Leisure activity		
Listening to music/radio		1378 (45.1%)
Reading newspaper/magazines		1464 (47.9%)
Going online (no interaction with other persons)		358 (11.7%)
Going online (interaction with other persons)		171 (5.6%)
Playing chess or cards		331 (10.8%)
Chatting with relatives/friends		2242 (73.4%)
Gardening (horticulture)		1279 (41.9%)
Taking a walk		2048 (67.1%)
Biking		816 (26.7%)
Outdoor fitness		572 (18.7%)
Group activity		482 (15.8%)

Abbreviations: CESD-10, the 10-item short form of the Center for Epidemiological Studies Depression Scale; TAAQ-SF, Taiwanese Attitude toward Aging Questionnaire Short Form.

**Table 2 ijerph-19-02654-t002:** Results of bivariate analysis: the association between attitude toward aging (ATA), and each demographic and health-related factor.

	n	Mean (SD) of ATA	*p*	Post Hoc	Eta Squared
**Demographic Factors**					
Age					
58–64	1254	42.6 (5.4)	0.000	All pairwise comparisons (*p* < 0.05)	0.10
65–74	954	40.3 (5.8)			
75–84	645	38.4 (6.2)			
85–98	201	37.1 (5.3)			
Gender					
Men	1543	41.2 (5.9)	0.000		0.009
Women	1511	40.0 (6.0)			
Education					
≤6 years	1912	39.6 (6.1)	0.000	All pairwise comparisons (*p* < 0.05)	0.06
7–12 years	803	41.9 (5.5)			
≥13 years	339	43.5 (5.1)			
Marital/cohabitation status					
Without spouse/partner	844	38.5 (6.2)	0.000		0.05
With spouse/partner	2164	41.5 (5.7)			
Financial satisfaction					
Dissatisfied	564	36.9 (6.7)	0.000	All pairwise comparisons (*p* < 0.05)	0.12
Acceptable	1165	40.3 (5.4)			
Satisfied	1324	42.5 (5.3)			
**Health-related Factors**					
Depression (CESD-10)					
Depressed (CESD-10 ≥ 10)	467	33.3 (5.8)	0.000		0.27
Not depressed (CESD-10 ≤ 9)	2585	42.0 (5.0)			
Physical function					
All 3 scales dependent	190	32.8 (6.0)	0.000	All pairwise comparisons (*p* < 0.05)	0.26
Any 2 scales dependent	526	36.7 (5.8)			
Any 1 scale dependent	474	39.8 (5.2)			
All 3 scales independent	1855	42.8 (4.8)			

Abbreviations: CESD-10, the 10-item short form of the Center for Epidemiological Studies Depression Scale.

**Table 3 ijerph-19-02654-t003:** Results of multiple regression analysis: factors significantly associated with attitude toward aging.

	β	t
Age (Reference: 58–64)		
65–74	−0.06	−4.04 ***
75–84	−0.10	−5.79 ***
85–98	−0.10	−6.05 ***
Gender (Reference: women)		
Men	0.02	−1.14
Education (Reference: ≤6 years)		
Education (7–12 years)	0.07	4.71 ***
Education (>12 years)	0.07	5.04 ***
Marital/cohabitation status (Reference: without spouse/partner)		
With spouse/partner	0.04	2.97 **
Financial status (Reference: dissatisfied)		
Average	0.13	6.81 ***
Satisfied	0.28	14.21 ***
Number of co-morbidities	−0.09	−5.97 ***
Depression (Reference: not depressed, CESD-10 ≤ 9)		
Depressed (CESD-10 ≥ 10)	−0.32	−20.82 ***
Physical function (Reference: independent in all 3)		
Any 1 scale dependent	−0.08	−5.69 ***
Any 2 scales dependent	−0.18	−10.68 ***
All 3 scales dependent	−0.23	−14.46 ***

Abbreviations: CESD-10, the 10-item short form of the Center for Epidemiological Studies Depression Scale; β, standardized coefficient. ** *p* < 0.01; *** *p* < 0.001.

**Table 4 ijerph-19-02654-t004:** The association of physical exercise, volunteer activities, and leisure activities with attitude toward aging controlling for significant demographic and health-related factors.

Variables	β	t
**Physical exercise** (Reference: no exercise) †		
Irregular	0.05	3.27 **
Regular	0.10	6.15 ***
**Volunteer activities** (Reference: no participation) †		
Volunteer service (participation)	0.07	5.23 ***
**Leisure activities** †		
Listening to radio/music	0.03	2.26 *
Reading newspapers/magazines/books	0.05	3.40 **
Going online (no interaction with other person)	0.01	0.40
Going online (interaction with other persons)	0.01	0.89
Playing chess or cards	0.02	1.75
Chatting with friends/relatives	0.04	2.85 **
Gardening (horticulture)	0.08	5.43 ***
Taking a walk	0.03	1.82
Bicycling	0.02	1.71
Outdoor fitness	0.07	4.63 ***
Group activity	0.06	3.96 ***

†: model adjusted demographic factors (age, education, marital and cohabitation status, and financial satisfaction) and health-related factors (numbers of co-morbidities, depression, and physical function). Abbreviation: β, standardized regression coefficient. * *p* < 0.05, ** *p* < 0.01, *** *p* < 0.001.

## Data Availability

TLSA data is a national study project, and its data is open to those who have been approved to use it for analysis (http://nhis.nhri.org.tw/ accessed on 13 October 2017).

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
