# Peer review of "Factors Associated with Attitudes toward Aging among Taiwanese Middle-Aged and Older Adults: Based on Population-Representative National Data"

_ijerph, 2022, doi:10.3390/ijerph19052654_

Round 1

Reviewer 1 Report

The paper is well written, the subject interesting and the method appropriate.

Paragraph of data analysis, line 125: I suggest to write the eta formula in a separate line or in note for a better reading of the text.

Author Response

We thank for this suggestion. We agree to make necessary change for a better reading of the text as long as it fit the requirement of typesetting.

Reviewer 2 Report

The reviewed manuscript concerns the important issue of attitude toward aging.

The study was conducted on a large, representative group of residents aged 58 and over. Statistical methods used for the analysis have been selected correctly.

For improved the quality of the manuscript, I would suggest the following comments:

  1. The study was based solely on residents of Taiwan. Is it not, then, an abuse to generalize the results to the entire Asian population as the title suggests?
  2. In the Introduction section, I propose to include a fragment describing the age structure of the residents of Taiwan - what percentage are middle-aged and older adults and what are the changes in this percentage over the last several dozen years? This will make readers aware of the problem of the aging of the Taiwanese population.
  3. I suggest rewording the Discussion section because its extensive fragments are a repetition of the results.

Author Response

We have revised the manuscript as recommended by the reviewer.

Reviewer 3 Report

The work requires improvement, it is unclear and does not bring anything innovative in such a presentation

  1. In the introduction, there is no information about the ATA occeny method, whether it is a questionnaire, objective - subjective assessment, who is making it. The authors mention "People with negative ATA ......" which means, what is the scale
    2. How were the people recruited for the study - did they sign the consent, what were the conditions for inclusion in the research group, how many people were included in the first study, and how many people resigned from continuing the study.
    3. Were the data collected and presented by the authors in 2011 the first study or a continuation? Since the program started in 1989 and was continued every 3-4 years, was the person examined in 1989 as a 58-year-old person tested and included in the group as 62 -66-70 .... years old ? Do the presented results only show the first study
    4. What were the questions in Taiwanese Attitude toward Aging Questionnaire Short Form, what scale do the authors use ... which means that the average result ranged between 11 and 55 is a good or bad result ....
    5. Showing the relationship between ATA and age is not new as it results from the aging process. It would be more interesting to show the difference between men and women, living in a larger - smaller city, addiction to education - that is what is missing. The authors treated the whole group as not different from each other, and they perceive the world differently by lonely people and those in a relationship, having a family .....
    5. In the current version, the work is a computation of the results, there is no attempt to explain what influences ATA

Author Response

We have revised the manuscript according to the reviewer's comments. 

Round 2

Reviewer 3 Report

The authors complied with my comments and improved the work